# Comparative Ubiquitome Analysis Reveals Deubiquitinating Effects Induced by *Wolbachia* Infection in *Drosophila melanogaster*

**DOI:** 10.3390/ijms23169459

**Published:** 2022-08-21

**Authors:** Qiong Zong, Bin Mao, Hua-Bao Zhang, Bing Wang, Wen-Juan Yu, Zhi-Wei Wang, Yu-Feng Wang

**Affiliations:** School of Life Sciences, Hubei Key Laboratory of Genetic Regulation and Integrative Biology, Central China Normal University, Wuhan 430079, China

**Keywords:** *Wolbachia*, *Drosophila melanogaster*, proteomics, deubiquitination, male fertility

## Abstract

The endosymbiotic *Wolbachia* bacteria frequently cause cytoplasmic incompatibility (CI) in their insect hosts, where *Wolbachia*-infected males cross with uninfected females, leading to no or fewer progenies, indicating a paternal modification by *Wolbachia*. Recent studies have identified a *Wolbachia* protein, CidB, containing a DUB (deubiquitylating enzyme) domain, which can be loaded into host sperm nuclei and involved in CI, though the DUB activity is not necessary for CI in *Drosophila melanogaster*. To investigate whether and how *Wolbachia* affect protein ubiquitination in testes of male hosts and are thus involved in male fertility, we compared the protein and ubiquitinated protein expressions in *D. melanogaster* testes with and without *Wolbachia*. A total of 643 differentially expressed proteins (DEPs) and 309 differentially expressed ubiquitinated proteins (DEUPs) were identified to have at least a 1.5-fold change with a *p*-value of <0.05. Many DEPs were enriched in metabolic pathway, ribosome, RNA transport, and post-translational protein modification pathways. Many DEUPs were involved in metabolism, ribosome, and proteasome pathways. Notably, 98.1% DEUPs were downregulated in the presence of *Wolbachia*. Four genes coding for DEUPs in ubiquitin proteasome pathways were knocked down, respectively, in *Wolbachia*-free fly testes. Among them, *Rpn6* and *Rpn7* knockdown caused male sterility, with no mature sperm in seminal vesicles. These results reveal deubiquitylating effects induced by *Wolbachia* infection, suggesting that *Wolbachia* can widely deubiquitinate proteins that have crucial functions in male fertility of their hosts, but are not involved in CI. Our data provide new insights into the regulatory mechanisms of endosymbiont/host interactions and male fertility.

## 1. Introduction

*Wolbachia* is a genus of endosymbionts that infect various arthropods and nematodes. The most common phenotype in insects induced by *Wolbachia* is cytoplasmic incompatibility (CI), where no or fewer progenies are produced when *Wolbachia*-infected males mate with uninfected females [1,2]. Recent studies have demonstrated that the host sperms are modified by *Wolbachia* Cif (CI factor) proteins and thus contribute to CI-defining embryonic lethality [3,4].

The ubiquitin proteasome pathway (UPP) is one of the most important proteolytic pathways in eukaryotic cells. It is involved in various biological processes, including cell cycle process, apoptosis, transcriptional regulation, DNA repair, and immune response [5]. Ubiquitination refers to a complex process where ubiquitin is first activated by ubiquitin-activating enzymes (E1), then transferred to ubiquitin-conjugating enzymes (E2), and finally linked to the target protein’s Lys residues by ubiquitin ligase enzymes (E3). Usually, the target proteins ligated to polyubiquitin chains at K48 lysine residues are transported to proteasome for proteolysis [5,6]. Before targeted proteins enter the proteolytic core of proteasomes, the ubiquitin should be removed normally by deubiquitinating enzymes (DUBs).

Ubiquitin was first found in the testes of trout and mammals [7], implying its essential role in male reproduction. A growing number of studies revealed that UPP plays a vital regulatory role in spermatogenesis [7,8,9,10]. Animal spermatogenesis is a complex process, including the maintaining and differentiation of germ stem cells (GSCs), mitosis, meiosis, and a complicated metamorphosis from round spermatids to unique morphology of mature sperms, which is called spermiogenesis. During spermiogenesis in *Drosophila melanogaster*, the round spermatid undergoes nuclear condensation, elongation of flagellar axoneme, mitochondrial morphogenesis, and extrusion of the excess cytoplasm [11]. Many of these processes are involved in UPP. Mutation of the polyubiquitin gene *Ubi-p63E* in *Drosophila* resulted in arrest at the G2/M transition of the first meiotic division [12]. During mouse spermiogenesis, H2A/H2B ubiquitination after meiosis may destabilize nucleosomes, and thus facilitate histone-to-protamine replacement [13,14]. Rnf8 E3 ubiquitin ligase can catalyze the ubiquitination of H2A and bind to H4K16 acetyltransferase MOF (males absent of the first) to promote H4 acetylation in mice [15]. Impairments of RNF8 and MOF-dependent histone ubiquitination and acetylation due to the absence of phosphorylated GRTH (Gonadotropin-regulated testicular RNA helicase) caused arrest of spermatogenesis at the round spermatid stage with defects in histone replacement, chromatin condensation and spermatid elongation [14,16]. In *Drosophila*, a lack of Archipelago, a member of the E3 ligase complex, also resulted in defects in nuclear shaping [17].

Studies have revealed that CidB (CifB), a *Wolbachia* protein including two nuclease domains and a DUB (deubiquitylating enzyme) domain, is deposited in maturing sperm and may act as a kind of “toxin” factor to poison the embryos when these sperm fertilize uninfected eggs [3]. A single mutation within the DUB domain of CidB reduced both DUB efficiency and CI strength [18]. However, a recent study showed that the inactivation of the deubiquitylase activity of CidB did not decrease the ability to induce CI-like phenotype in *Drosophila*. The deubiquitylase activity of CidB could contribute to the localization or stabilization of CidB in the spermatid after histone-to-protamine transition and could thus be involved in CI [3]. Shropshire et al. demonstrated that *cifB* expression could not explain age-dependent CI-strength variation. The expression level of *cifB* in testes increases as *w*Mel-infected males’ age, though CI strength declines [19]. By comparative proteomics of the spermatheca and seminal receptacle (SSR, containing sperm proteins and seminal fluid proteins from their mates) from uninfected females shortly after mating with *Wolbachia*-infected or uninfected males, we previously identified several proteins in UPP that were significantly downregulated due to *Wolbachia* infection [20], consistent with the role of UPP in *Wolbachia*-induced modification in host sperms. These indicate that the role of deubiquitylase activity of *Wolbachia* proteins in CI remains controversial.

To investigate whether and how *Wolbachia* modifies host sperms through UPP, we compared the global profiling of the proteome and ubiquitome of *Wolbachia*-infected (DWT) and uninfected *D. melanogaster* testes (DTT). Surprisingly, 98.1% of differentially expressed ubiquitinated proteins (DEUPs) were downregulated in the presence of *Wolbachia*, supporting the deubiquitinating function of *Wolbachia* proteins. RNAi analysis demonstrated that some of the DEUPs were essential for *Drosophila* spermatogenesis. Our data provide not only new evidences supporting the deubiquitinating role of *Wolbachia* proteins in host testes, but also an abundant dataset for further studies on their functions in male fertility of animals.

## 2. Results

### 2.1. Proteomic Profiles of Wolbachia-Infected and Uninfected Drosophila Testes

To investigate the protein background of the testes of *D. melanogaster*, and the effect of *Wolbachia* infection on protein expressions in fly testes, we first performed LC-MS/MS to identify total proteins in the testis of 1-day-old (1 d) flies with or without *Wolbachia*. In total, 1038 proteins and 1185 proteins were quantified in DWT and DTT, respectively (Figure 1A, Appendix A).

### 2.2. Analysis of Differentially Expressed Proteins between DWT and DTT

Among the quantified common proteins in both the DWT and DTT groups, differentially expressed proteins (DEPs) were selected with the criteria of a fold change of ≥1.5 and a *p*-value of <0.05. On the base of these criteria, 148 proteins were downregulated and only 5 proteins were upregulated in DWT relative to DTT. Furthermore, 320 proteins were specifically detected in DTT, and 170 were specifically in DWT.

Based on the Gene Ontology (GO) biological process (BP), we found that, among the 468 proteins that were not detected and downregulated (320 + 148) in DWT relative to DTT (integrally regarded as downregulated thereafter), the largest group of proteins was related to peptide biosynthetic process, followed by homeostatic process, purine-containing compound metabolic process, generation of precursor metabolites and energy, and ribonucleoprotein complex biogenesis. Proteins involved in regulation of catabolic process, regulation of protein stability, and mitochondrial transport were also significantly enriched (Figure 1B). Pathway enrichment of the downregulated proteins showed that many proteins were involved in metabolic pathways, ribosome, oxidative phosphorylation, RNA transport, and citrate cycle (Figure 1C).

Among the proteins that were specifically identified and upregulated (170 + 5) in DWT relative to DTT (integrally regarded as upregulated thereafter), the majority of proteins were involved in peptide biosynthetic process, organic acid metabolic process, small molecule biosynthetic process, purine-containing compound metabolic process, and generation of precursor metabolites and energy (Figure 1D). Pathway analyses of these proteins exhibited that metabolic pathways, post-translational protein modification, protein processing in endoplasmic reticulum, translation, RNA transport, and proteasome were significantly enriched (Figure 1E).

To explore how the DEPs might interact with diverse pathways, we constructed the protein–protein interaction (PPI) networks that were involved in the ubiquitin-proteasome pathway and in male fertility (Figure 2). A network containing 24 DEPs, including 14 downregulated and 10 upregulated proteins, was retrieved to be associated with UPP, including subunits of the regulatory particle lid of proteasome Rpn1, Rpn2, Rpn12R, and subunits of the core particle of proteasome Prosβ6 (Figure 2A). A large network consisting of 57 DEPs, including 45 downregulated and 12 upregulated proteins, was constructed to be associated with male fertility (Figure 2B). bgcn, CadN, Nedd8, and Tctp were all involved in testis germline stem cell maintenance and differentiation, while loopin-1, S-Lap2, S-Lap3, and S-Lap8 were all in the major mitochondrial derivative and related to the formation of sperm nebenkern.

### 2.3. Proteome-Wide Analysis of Ubiquitinated Proteins in DWT and DTT

To investigate the global effect of *Wolbachia* infection on protein ubiquitination in fly testes and thus on male fertility, we performed proteome-wide analysis of lysine ubiquitination sites and proteins in DWT and DTT.

A total of 471 lysine ubiquitination sites were identified in 356 ubiquitinated peptides, corresponding to 600 lysine ubiquitinated proteins (Figure 3A, Appendix A). The length of most ubiquitinated peptides ranged from 6 to 47 amino acids. Among these 356 ubiquitinated peptides, most (275, 77.25%) of them contained only a single ubiquitination site, while the others carried multiple ubiquitination sites, including 8 (2.25%) with five or more sites (Figure 3B). Hsc70-4 was the most intensely ubiquitinated protein, having 8 different lysine ubiquitination sites (Figure 3C).

### 2.4. Analysis of Differentially Expressed Ubiquitinated Proteins (DEUPs) between DTT and DWT

When the ubiquitome quantification results were normalized to protein amount in each sample, we found that 14 proteins were significantly downregulated but no proteins were significantly upregulated in DWT relative to DTT. In addition, 289 ubiquitinated proteins were specifically identified in DTT, and six were specifically in DWT. Among the ubiquitinated proteins that were not expressed and downregulated (289 + 14) in DWT relative to DTT (integrally regarded as downregulated thereafter in DWT, i.e., specifically or highly expressed in DTT), a large group of ubiquitinated proteins for BP classification were related to the carboxylic acid metabolic process, generation of precursor metabolites and energy, cytoplasmic translation, and proteasome-mediated ubiquitin-dependent protein catabolic process. Proteins involved in chromatin assembly or disassembly were also significantly enriched (Figure 4A). KOBAS pathway analyses revealed that many downregulated ubiquitinated proteins were highly enriched in carbon metabolism, ribosome, and proteasome pathways (Figure 4B).

There were only six ubiquitinated proteins that were specifically identified in DWT (regarded as upregulated thereafter), including fan, Atpalpha, ND-20, SH3PX1, G6P, and Cse1. GOBP analyses showed that fan was involved in sperm individualization [21]. Atpalpha and ND-20 were associated with proton transmembrane transport [22] and determination of adult lifespan [23]. G6P was related to gluconeogenesis [24]. Cse1 was involved in protein export from the nucleus [25]. ND-20, fan, and G6P were significantly enriched in the metabolism pathway.

A network containing 24 DEUPs was involved in the ubiquitin–proteasome pathway, including ubiquitin-activating enzyme Uba1, ubiquitin-conjugating enzyme Ubc4, and many subunits of the proteasome, such as Rpn1, Rpn7, and Prosα7, Prosβ6. All of them were significantly downregulated in the presence of *Wolbachia* (Figure 5A). A large network containing 46 DEUPs was related to male fertility (Figure 5B). For example, bel has been shown to play an essential role in male germline stem cell maintenance and division in *Drosophila* [26]. S-Lap5, S-Lap7, and S-Lap8 were all the main components of the major mitochondrial derivative (eventually form the nebenkern) of sperms [27,28]. These suggest that UPP and pathways involved in male fertility displayed dense protein interaction networks which may lead to their coordination and cooperation in male reproduction.

### 2.5. The Correlation Analysis between DEPs and DEUPs

To assay the correlation between DEPs and DEUPs, 468 downregulated DEPs and 303 downregulated DEUPs were used to graph a Venn diagram. As shown as Figure 6A, 81 downregulated proteins were overlapped, including 13 proteins that are ribosomal proteins involved in protein synthesis, such as RpL4, and RpS27, and 4 proteins that are in the UPP, such as proteasome subunits Rpn1, Rpn2, and Rpn12R. This indicates that the pathways involved in protein synthesis and degradation through UPP in fly testis are notably affected by *Wolbachia* infection.

However, 222 out of a total 303 DEUPs (66.67%) were not found in the DEPs (Figure 6A). For example, bol (boule), which plays an essential role in spermatogenesis for meiosis and spermatid differentiation [29,30], was not identified in DWT (regarded as downregulated) in ubiquitination expression, while there was no significant change in the protein level (1.18-fold) between DWT and DTT. On the other hand, 387 out of 468 (82.69%) DEPs were not found in DEUPs. For instance, the S-Lap2 was significantly downregulated (0.35-fold, *p* < 0.05) in the protein level in the presence of *Wolbachia*, but did not show a difference in DEUPs (0.68-fold, *p* > 0.05). KOBAS analyses of these overlapped 81 proteins showed that most of the proteins were involved in metabolic pathways, ribosome, oxidative phosphorylation, and the proteasome pathway (Figure 6B).

Nevertheless, only three upregulated proteins were overlapped between DEPs and DEUPs, including Cse1, ND-20, and SH3PX1 (Figure 6C). Ubiquitinated Atpalpha, fan, and G6P were all specifically identified in DWT (regarded as upregulated), while in the protein level Atpalpha and fan were not significantly different between DWT and DTT, G6P was downregulated (0.40-fold) in the presence of *Wolbachia* though the *p*-value was > 0.05.

### 2.6. Quantitative Reverse Transcription Polymerase Chain Reaction (qRT-PCR) Validation and Screening

Based on a larger difference in expression in the proteome and ubiquitome analyses, 12 protein coding genes were selected for qRT-PCR for further screening candidates involved in male fertility. As shown in Figure 7, nine genes including *Rpn12R*, *His2Av*, *Rpn7*, *Prosα7*, *CSN4*, *Phb2*, *Prosβ1*, *Cul2*, and *CSN5* were all significantly downregulated due to *Wolbachia* infection in the testes of *D. melanogaster*, exhibiting similar changes as in the mass spectrometric analyses. Both Rpt4R and Rpn6 are important UPP members and exhibited lower expression levels in proteomic analyses in DWT than in DTT, despite a higher *p*-value (>0.05). Hsc70-4 had the most ubiquitination sites (Figure 3C), and showed much lower expression levels (0.19-fold) in ubiquitome analyses in the presence of *Wolbachia*, although with a *p* > 0.05. Therefore, the expressions of these three protein coding genes were also measured by qRT-PCR, and the result showed that these three genes were significantly downregulated in fly testes by *Wolbachia* infection (Figure 7).

### 2.7. Functional Analysis of Candidates in Male Fertility

We next used gene knockdown to test some genes coding for four subunits of proteasomes, including *Rpn6*, *Rpn7*, *Rpt4R*, and *Pros**α7*, for their functions in male fertility. qRT-PCR analyses showed that these genes were all successfully knocked down driven by bamGal4 in fly testes (Figure 8A).

Then, the 1 d gene knockdown males were selected to respectively cross with *Wolbachia*-free virgin females. The hatch rates of eggs derived from these crosses were all significantly lower than that from the control group (Figure 8B). Especially, the *Rpn6* and *Rpn7* knockdown males were completely sterile, the egg hatch rate from these two cross groups was 0 (Figure 8B).

To examine whether the knockdown of *Rpn6* and *Rpn7* in fly testes damaged spermatogenesis, thus causing male sterility, we dissected *Rpn6* or *Rpn7* knockdown fly testes, respectively, and stained them with Vasa antibody and DAPI. We observed that the testis from *Rpn6* knockdown males did not show visible differences in the whole appearance when compared to the control testes (Figure 8C,G). In the head region, the control testes contained developing germ cells with different sizes (Figure 8D), while in the *Rpn6* knockdown testes, we observed some tumor-like cysts containing many small cells that could not be stained by Vasa antibody (arrows in Figure 8H). DAPI staining showed that in the base of the control testes, the spermatid nuclei were tightly clustered (arrows in Figure 8E). In contrast, the spermatid nuclei in *Rpn6* knockdown testes were dissociated from the bundles, with some scattered around individually, and some accumulated together loosely (arrows in Figure 8I). When observing carefully, we found that the nuclei of the spermatid were less condensed (Figure 8I) than those in the controls (Figure 8E). Finally, the seminal vesicle was completely empty (Figure 8J), in contrast to the vesicle full of mature sperms with needle-like heads in the control (Figure 8F). Similarly, the seminal vesicle of *Rpn7* knockdown testes was also empty (Figure 8N). In addition, *Rpn7* knockdown testes looked smaller than the controls (Figure 8C,K), and the defects in germ cell developments occurred earlier since no big spermatocytes and spermatids appeared at all (Figure 8L,M).

To investigate whether these genes were involved in CI induced by *Wolbachia*, we used *Wolbachia*-infected females (Dmel *w*Mel) to cross with the gene knockdown males. The results showed that the *Wolbachia*-infected females could not rescue *Rpn6* or *Rpn7* knockdown in male flies, the egg hatch rates were still both 0%. The hatch rate of eggs from the cross group with *Prosalpha7* knockdown males and Dmel *w*Mel females increased only slightly from 56.43% to 67.45%, although the statistical analyses showed that the difference was significant (Appendix A). This indicates that these genes are essential for male fertility but they are not involved in *Wolbachia*-induced CI.

## 3. Discussion

In this study, we provide a first comprehensive quantitative analysis of the protein and protein ubiquitination differences between DWT and DTT. The current workflow led us to identify 643 DEPs, including 468 downregulated and 175 upregulated proteins, and 309 DEUPs, with 303 downregulated and 6 upregulated ubiquitinated proteins in the presence of *Wolbachia*. Many of them are involved in UPP and male fertility. Over 98% of DEUPs were downregulated due to *Wolbachia* infection, supporting a deubiquitinating role of proteins provided by *Wolbachia*.

Most proteins in the testis of *D. melanogaster* were downregulated by *Wolbachia.* We have previously identified differentially expressed proteins in SSR of uninfected females mated with 1 d *Wolbachia*-infected and uninfected male flies, and found that most (71.08%) of them were downregulated [20]. In the present study, we identified 643 DEPs, and again, most (72.78%) were downregulated due to *Wolbachia* infection. Among those significantly changed proteins, many of them are involved in metabolic processes, which is similar to the results observed in our previous study [20,31] and reports from other groups [32,33]. In studies on human, mouse, boar, and rainbow trout sperm proteome, many sperm proteins were also related to the metabolic processes [34,35,36,37]. Many DEGs were involved in ribosome and proteasome pathways. RpL22 and RpL22-like have been demonstrated to be essential for spermatogenesis [38,39]. Proteasome has been shown to play a critical role in spermatogenesis and fertilization [9,40,41]. Around a third of the 32 proteasome subunits have evolved to be testis-specific, and been detected to predominantly localize to the nucleus of mature, motile sperm [42,43]. Mutation of these testis-specific proteasome subunits resulted in severe defects in meiosis, nuclear maturation, and sperm individualization [41,43]. Yu et al. have demonstrated that protein synthesis and degradation are essential to regulate GSC self-renewal or differentiation [44]. This is likely because, during spermatogenesis, protein homeostasis should be precisely controlled to ensure the functions of certain proteins at a definite stage and the elimination of them after this stage, and finally, the sperm must accumulate some proteins preparing for the generation of enough energy for upcoming strenuous movement and fertilization. Furthermore, sperm mitochondria are normally labelled through ubiquitination during spermatogenesis [45] and thus marked for elimination by the proteasome complex after entering the egg. Prohibitins have been demonstrated to play a role in mtDNA inheritance [46] and are targets for ubiquitination in sperm mitochondria [47]. The species-specific expression differences of prohibitin coding genes have been suggested to be involved in reproductive isolation [48]. In this study, Phb2 was significantly downregulated by *Wolbachia* infection. In CI embryos, the *Wolbachia*-modified sperm cannot normally develop when entering an uninfected egg, which has been suggested to contribute to reproductive isolation in insects [1,49,50,51]. Hence, these differentially expressed proteasome subunits and prohibitin might play a part in CI induced by *Wolbachia*. Future analyses will be needed to disentangle the contributions of eliminating paternal mitochondria (nebenkern) to CI induced by *Wolbachia*.

The ubiquitome difference between DWT and DTT exhibits a deubiquitinating role of proteins from *Wolbachia.* Here, we globally compared the ubiquitome of DWT and DTT. We identified many more downregulated ubiquitinated proteins than upregulated proteins (303 vs. 6) due to *Wolbachia* infection. Notably, there were up to 289 ubiquitinated proteins that were specifically identified in DTT, but could not be detected in DWT, including His2Av and His2B. H2A/H2B ubiquitination is required for histone-to-protamine replacement during sperm nuclear condensation [11]. A recent study has suggested that the deubiquitylase activity of CidB could contribute to the localization or stabilization of CidB in the spermatid after histone-to-protamine transition [3]. Expressions of CifA and CifB proteins cause abnormal histone retention and protamine deficiency in host sperms, thus modifying paternal genome integrity, and result in embryonic lethality [4]. Therefore, our result reveals the deubiquitinating effects of *Wolbachia* proteins in the testis of *D. melanogaster*, and may thus damage male fertility. Furthermore, eIF3h, recently being demonstrated to be a novel deubiquitinase [52], was identified specifically in DWT. This may also result in the wide deubiquitination of proteins in DWT.

Up to 24 DEUPs involved in UPP were downregulated due to *Wolbachia* infection. For example, Rpn7 and Prosα7 were unable to be detected in DWT (Figure 6A), although they were indeed expressed in DTT. As discussed above, proteasomes play a pivotal function during spermatogenesis and fertilization in many animals, including mammals [43,53,54,55,56]. These suggest that many subunits of proteasome were deubiquitinated due to *Wolbachia* infection, which will alter the functions of UPP and impair male fertility.

Heat shock protein cognate 4 (Hsc70-4) protein was found to contain eight different lysine ubiquitination sites (Figure 3C). The ubiquitinated Hsc70-4 was only 0.19-fold in DWT relative to DTT. Hsc70-4 was required for the self-renewal of germline stem cells and differentiation of spermatogonia in *Drosophila.* Knockdown of *Hsc70-4* resulted in fly testes full of small undifferentiated germ cells [44]. Recent studies suggested that Hsc70-4 possibly targeted Akt or Pdk1 acting downstream of PI3K, thus regulating spermatocyte growth and meiosis initiation [57]. Therefore, the dramatic reduction of ubiquitinated Hsc70-4 by *Wolbachia* infection might damage germ cell differentiation in *Drosophila* testes.

DEPs and DEUPs include new regulators of spermatogenesis. Knockdown of four candidates all resulted in significantly reduced male fertility, with *Rpn6* and *Rpn7* giving no progenies. Rpn6 was also identified to be downregulated by *Wolbachia* infection in SSR [20]. The mutation of *Rpn6* in flies was able to hatch, but never develop to adults, indicating an essential role in *Drosophila* development [58]. In this study, we found that knockdown of *Rpn6* in fly testes did not affect the viability, but caused male sterility. In the anterior region of the testes, there appeared some tumor-like cysts containing small cells that were negative for Vasa, indicating that they were not germ cells. Further studies will be necessary to determine what kind of cells are in these tumor-like cysts. In the basal region, the spermatid nuclear bundles were disrupted, and scattered spermatids contained less condensed nuclei. These suggest that Rpn6 is required for spermiogenesis. *Rpn7* knockdown in fly testes led to viable adults, but resulted in defects in the earlier stage of spermatogenesis, with no bigger spermatocytes in testes, indicating a crucial role of Rpn7 in cell growth before meiosis. However, the *Wolbachia*-infected females could not rescue the defects in male fertility, suggesting that these genes are indeed required for spermatogenesis, but they are not involved in *Wolbachia*-induced CI.

In conclusion, we performed proteome-wide analysis of ubiquitylated proteins in DWT and DTT, and identified 643 DEPs and 309 DEUPs. Most of the DEPs and DEUPs are involved in protein metabolism and male fertility. Notably, 98.1% of the DEUPs were downregulated in the presence of *Wolbachia*. Knockdown of genes coding for DEUPs involved in UPP in the testis significantly decreased the male fecundity. Of those, *Rpn6* and *Rpn7* are essential for *Drosophila* spermatogenesis. Our findings reveal a deubiquitinating role of proteins from *Wolbachia,* thus influencing host male fertility, and provide insights into the mechanisms of both endosymbionts/hosts interactions and male animal fertility.

## 4. Materials and Methods

### 4.1. Fly Lines

The *D. melanogaster* infected *w*Mel *Wolbachia* was a gift from Professor Scott O’Neill (Monash University, Melbourne, Australia). *Wolbachia*-uninfected flies were generated by tetracycline treatment according to established protocols [59] and confirmed to be *Wolbachia*-free by polymerase chain reaction (PCR) with *Wolbachia* surface protein gene (wsp) primers (Table 1). To eliminate the influence of residual tetracycline, the flies were reared in normal (tetracycline free) medium for more than 6 generations. The transgenic RNAi lines, including Rpt4R-hp (65361) and Prosα7-hp (67909), were obtained from the Bloomington Drosophila Stock Center (Bloomington, IN, USA). Rpn6-hp (THU4060) was from the Tsing Hua Fly Center, Beijing, China. Rpn7-hp (v22104) was from the Vienna Drosophila Resource Center (Vienna, Italy). The bamGal4 vp16 line was a kind gift from Professor Zhaohui Wang at the Institute of Genetics and Developmental Biology, Chinese Academy of Sciences. All flies were raised in 150 mL conical flasks with standard corn/sugar medium at 25 °C.

### 4.2. Protein Extraction

The testes of 1 d male flies were dissected. Then, testes were grinded three times in protein lysis buffer (8 M urea, 50 mM NH_4_HCO_3_, and 10% Protease Inhibitor) and centrifuged (12,000× *g* at 4 °C for 20 min) to remove the debris. The supernatant was collected and the protein concentration was determined (Appendix A) using the BCA protein Assay Kit (Beyotime, Shanghai, China).

### 4.3. Acetone Precipitation and Trypsin Digestion

For each biological repeat, 100 µg testis protein (from around 60 1 d male flies) was precipitated in 600 µL cold acetone for 2 h at −20 °C. The precipitated protein was obtained by centrifuging at 14,000× *g* at 4 °C for 10 min. Then, the protein solution was reduced with 10 mM dithiothreitol for 30 min at 37 °C and alkylated with iodoacetamide (11 mM) for 30 min at 37 °C in the dark. The sample was then diluted by adding 50 mM NH_4_HCO_3_ to urea concentration less than 2 M. Finally, the sample was digested by trypsin (Promega, Chilworth, UK) at 37 °C overnight with an enzyme to substrate ratio of 1:50. The digestion was stopped by adding 2 μL formic acid. After trypsin digestion, proteins were desalted by the stage-tip method [60] and analyzed by LC-MS/MS, respectively. As the initial total amounts of proteins in each group were consistent, the peak area of protein was used to represent the relative expression level of protein.

### 4.4. Affinity Enrichment of the Ubiquitinated Protein (IP)

To enrich ubiquitinated peptides, 100 μL of Protein A MagBeads (GenScript, Nanjing, China) were rinsed with 1 mL wash buffer (20 mM Na_2_HPO_4_, 0.15 M NaCl, pH = 7.0) twice. Next, the beads were fully suspended in 1 mL binding buffer (20 mM Na_2_HPO_4_, 0.15 M NaCl, pH 7.0). Two microliters of ubiquitin antibody (rabbit anti-ubiquitin polyclonal antibody, Bioss, MA, USA, catalog: bs-1549R) were added to the suspension and gently mixed for 40 min at room temperature (RT) to obtain the anti-ubiquitin beads. Then, 2 mg of testes protein (from around 1200 1 d male flies) were incubated with anti-ubiquitin beads at RT for 1 h with gentle shaking. The beads were washed 3 times with 1 mL PBS. The bounded protein was eluted from the beads with 100 μL elution buffer (0.1 M glycine, pH 2.5) twice. Finally, the eluted fractions were combined and increased pH to about 8 using 20 μL of Neutralization buffer (1 M Tris, pH 8.5). The subsequent acetone precipitation and trypsin digestion were performed as described above.

### 4.5. LC-MS/MS Analysis

The Exactive™ Plus Orbitrap high-resolution mass spectrometry and the EASY-nLC™ 1200 system (Thermo Fisher Scientific, Waltham, MA, USA) were used for analyzing protein samples. A Homemade C18 column (3 μm, 100 Å pores, 15 cm length, 75 μm diameter) was made for separating the sample by a flow rate of 0.5 μL/min. The linear gradient of 5–80% flow phase B for 120 min was used for peptide segment separation (flow phase A: 100% of water and 0.1% formic acid; flow phase B: 100% acetonitrile, 0.1% formic acid). Data-dependent acquisition (DDA) was used for mass spectrometry with a m/z scan range of 350–2000 m/z (resolution 70,000), automatic gain control (AGC) was set at 3 × 10^6^, peptides were detected at a resolution of 70,000, HCD fragmentation with a spectral resolution of 7500, normalized collision energy was 27%, dynamic exclusion time was 40 s.

### 4.6. Database Search and Bioinformatics Analysis

The resulting data from mass spectrometry were processed by using Proteome Discoverer 2.1. The UniProt protein database subset of *D. melanogaster* was used for searching as the following settings: Cleavage enzyme was trypsin, missing cleavage up to 2, fixed modification was carbamidomethylation of cysteines, GG[K] was variable modification, maximum mass deviation for parent ions was 10 ppm, maximum mass deviation for fragments was 0.02 Da, and false detection rate (FDR) was <1%.

*Drosophila* proteome from UniProt (https://www.uniprot.org/, accessed on 30 March 2021) was set as the reference dataset. The ubiquitinated protein levels were normalized to the corresponding protein amount in each sample. Two-sample *t*-test was done to compare differences between DWT and DTT. Among the quantified proteins, DEPs and DEUPs were selected with the criteria of a fold change of ≥1.5 and a *p*-value of <0.05.

Enrichment analysis of GO and pathway was performed respectively in WebGestalt (http://www.webgestalt.org/, accessed on 2 May 2022) [61] and KOBAS online software (http://kobas.cbi.pku.edu.cn/, accessed on 2 May 2022) [62], and visualized by R 4.2.0 (https://cran.r-project.org/, accessed on 2 May 2022). The PPI network among the surveyed proteins was constructed using the STRING database (version 11.5), and Cytoscape (v3.9.0) was used for data visualization. Venn diagrams were drawn by online software Bioinformatics (https://bioinformatics.psb.ugent.be/webtools/Venn/, accessed on 9 May 2022).

### 4.7. qRT-PCR

DWT and DTT of 1 d flies were respectively dissected into TRIzol reagent (Invitrogen) for total RNA extraction. The cDNA was synthesized from 2.5 μg total RNA using TransScript^®^ One-Step gDNA Removal and a cDNA Synthesis SuperMix kit (TransGen Biotech, Beijing, China). Specific primers for tested genes were designed based on sequences from the flybase database (Table 1). QPCR was performed on a CFX Connect^TM^ Real-Time PCR Detection System (Bio-Rad) with ChamQ Universal SYBR qPCR Master Mix (Vazyme, Nanjing, China). The qPCR program setting was as follows: 95 °C for 3 min, followed by 40 cycles of 95 °C for 10 s, 60 °C for 30 s, and a melting curve was constructed from 65 °C to 95 °C with increment 0.5 °C for 5 s. *rp49* was used as the reference gene, and relative expression of the tested gene against *rp49* was calculated using 2^−ΔΔCT^ [ΔΔC_T_ = (C_T,target_ − C_T,rp49_)_experiment_ − (C_T,target_ − C_T,rp49_)_control_] [63]. Three biological replicates and 3 technical replicates for each biological replicate were performed for this experiment.

### 4.8. Fertility Test

bamGal4 virgin females were used to cross with RNAi male flies to generate flies with gene knockdown in early spermatogenesis. Flies from the cross of bamGal4 females and *w^1118^* males (*bamGal4 > w*-) were used as the control. Fifteen 1 d gene knockdown males were arranged to cross with 10 females (3~5-day-old). After mating for around 12 h, all males were removed. Eggs were then collected and incubated at 25 °C for 48 h. Hatch rates were determined by counting the number of hatched eggs to total eggs. At least three biological repeats per cross type were performed.

### 4.9. Immunofluorescent Staining

About 30 testes were dissected and fixed with 4% paraformaldehyde at RT for 30 min, and washed with PBS 3 times (5 min each). After permeabilizing with PBST (PBS + 0.1% Triton X-100) for 30 min, the testes were washed and blocked with 5% goat serum at RT for 1 h. Then, the testes were incubated with rat anti-Vasa (1:200, DSHB, Lowa, IA, USA, cat No. AB760351) overnight at 4 °C. After washing, samples were incubated with Dylight 594, goat anti-rat IgG (1:200, Abbkine, cat No. A23440) in darkness for 3 h. After three rinses with PBST (10 min each), samples were mounted to slides using an antifading medium containing 2 μg/mL 4′-6-diamidino-2-phenylindole (DAPI) (Solarbio, Beijing, China). The slides were observed and photographed using a Leica SP8 Laser confocal microscopy (Leica, Wetzlar, Germany) and images were processed with ImageJ.

### 4.10. Statistical Analysis

Results were expressed as mean ± SE. Student *t*-test was used to analyze the difference between mean values. *p* < 0.05 was considered as significant difference.

## Figures and Tables

**Figure 1 ijms-23-09459-f001:**
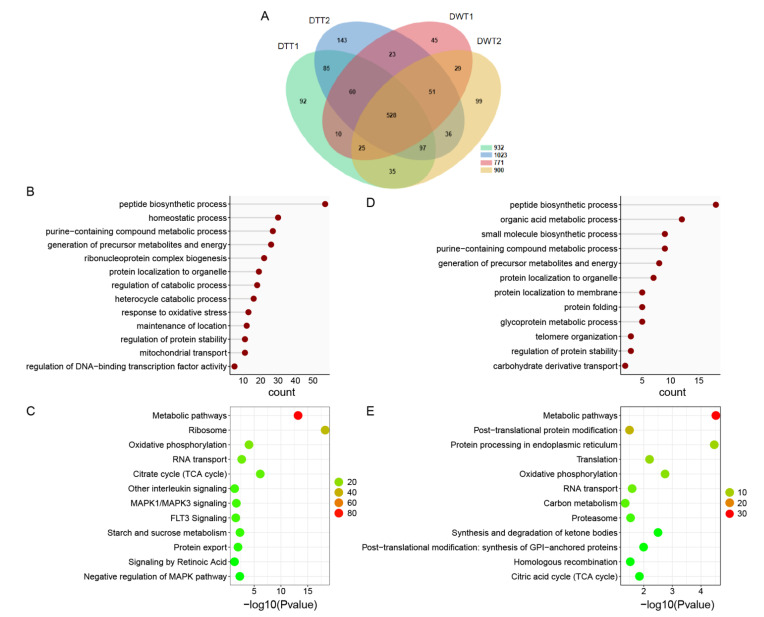
The proteome analysis of DWT vs. DTT. (**A**) Venn diagram showing overlaps of protein in DWT and DTT. (**B**) Gene ontology enrichment analysis based on biological process (GOBP) of proteins that were downregulated in the presence of *Wolbachia*. (**C**) KOBAS pathway analysis of proteins that were downregulated due to *Wolbachia* infection. (**D**) GOBP analysis of proteins that were upregulated in the presence of *Wolbachia*. (**E**) KOBAS pathway analysis of proteins that were upregulated due to *Wolbachia* infection. DWT: *Drosophila* (*Wolbachia*-infected) testes; DTT: *Drosophila* (treated with tetracycline, *Wolbachia*-free) testes. (**C**,**E**): The color indicates the difference in protein counts. The redder the color, the more the count; the greener the color, the less the count.

**Figure 2 ijms-23-09459-f002:**
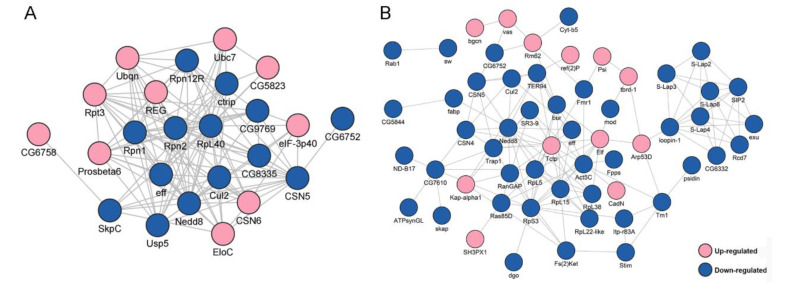
Protein–protein interaction (PPI) analyses of differentially expressed proteins (DEPs) between *Wolbachia*-infected and uninfected testes of *D. melanogaster*. (**A**) PPI analyses of DEPs that were involved in the ubiquitin-proteasome pathway. (**B**) PPI analyses of DEPs that were associated with male fertility.

**Figure 3 ijms-23-09459-f003:**
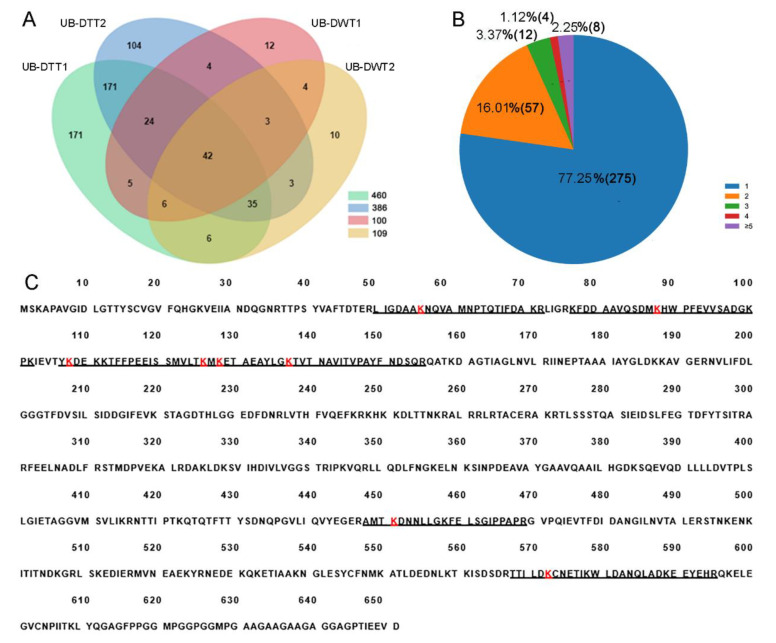
Identification of lysine ubiquitination sites in DTT and DWT. (**A**) Venn diagram showing overlaps of proteins in UB-DTT and UB-DWT. (**B**) Distribution of the number of lysine ubiquitination sites in the identified ubiquitinated peptides. (**C**) Eight lysine ubiquitination sites (red K) present in HSC70-4 protein. The underline sequences indicate the detected peptides. The red “K” is the lysine ubiquitination site.

**Figure 4 ijms-23-09459-f004:**
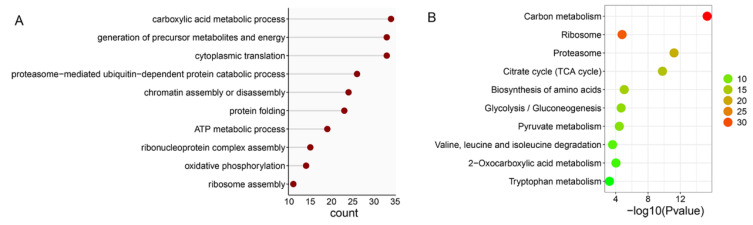
Analysis of downregulated ubiquitinated proteins in the testis of *D. melanogaster* due to *Wolbachia* infection. (**A**) Gene ontology enrichment analysis based on biological process. (**B**) KOBAS pathway analysis. The color indicates the difference in protein counts. The redder the color, the more the count; the greener the color, the less the count.

**Figure 5 ijms-23-09459-f005:**
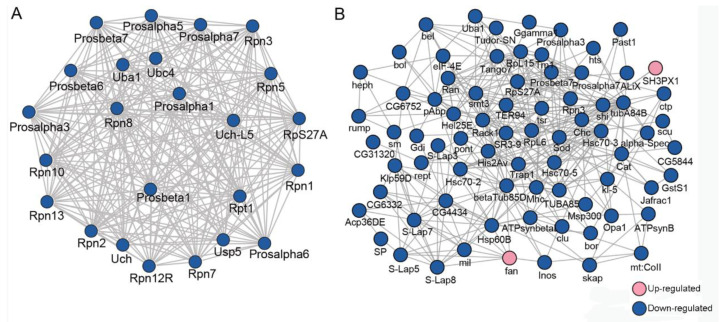
PPI analyses of DEUPs. (**A**) PPI analyses of DEUPs that were involved in the ubiquitin–proteasome pathway. (**B**) PPI analyses of DEUPs that were associated with male fertility.

**Figure 6 ijms-23-09459-f006:**
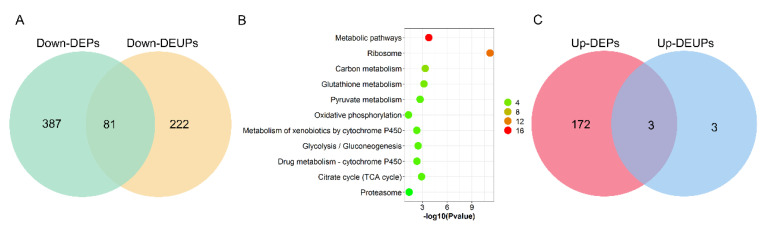
The correlation analysis between differentially expressed proteins and differentially expressed ubiquitinated proteins due to *Wolbachia* infection in the testis of *D. melanogaster*. (**A**) Venn diagram of the correlation numbers between downregulated proteins and downregulated ubiquitinated proteins. (**B**) GOBP enrichment of overlapped downregulated proteins and downregulated ubiquitinated proteins. The color indicates the difference in protein counts. The redder the color, the more the count; the greener the color, the less the count. (**C**) Venn diagram of the correlation numbers between upregulated proteins and upregulated ubiquitinated proteins.

**Figure 7 ijms-23-09459-f007:**
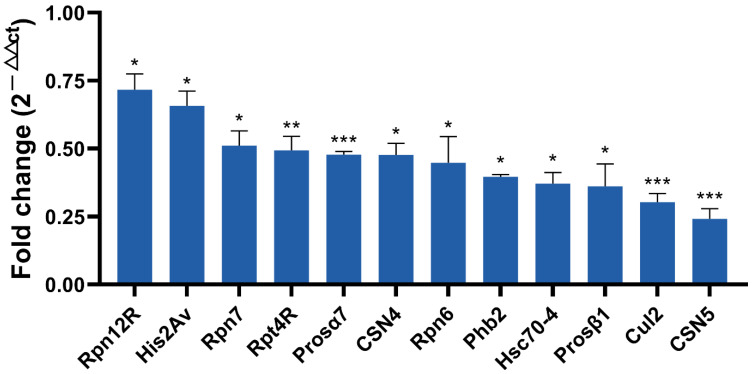
qRT-PCR validation of selected differentially expressed protein coding genes identified by proteomic and ubiquitome analyses. * *p* < 0.05; ** *p* < 0.01; *** *p* < 0.001.

**Figure 8 ijms-23-09459-f008:**
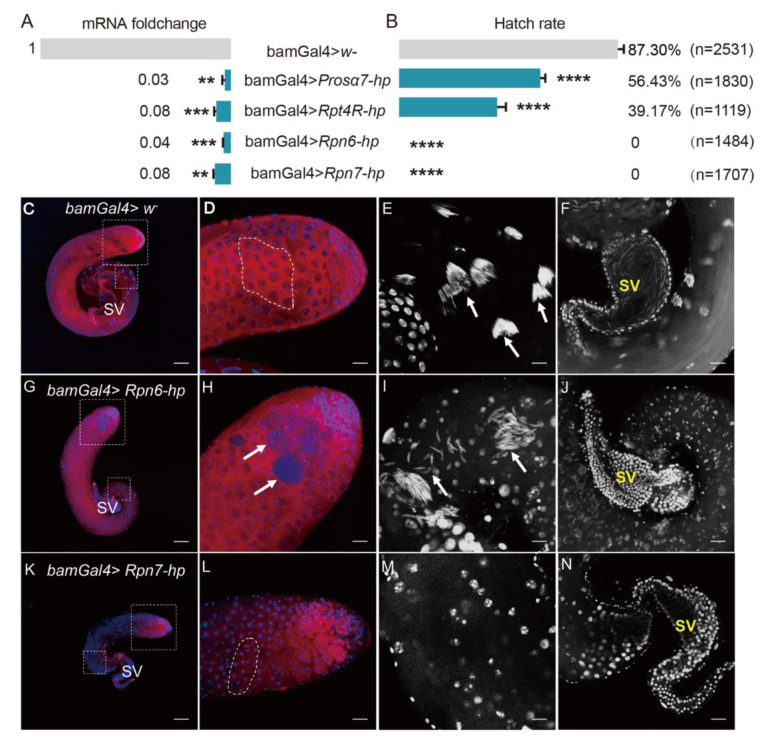
Knockdown of candidate genes in the testis of *D. melanogaster* impaired male fertility. (**A**) qRT-PCR analyses showed that the four genes were successfully knocked down driven by bamGal4 in fly testes. (**B**) Hatch rates of eggs derived from the crosses with the four gene knockdown males. bamGal4 *> w*– males were used as control. ** *p* < 0.01; *** *p* < 0.001; **** *p* < 0.0001. (**C**–**N**) Fly testes stained with Vasa antibody and DAPI. (**C**–**F**) Control testis; (**G**–**J**) *Rpn6* knockdown testis; (**K**–**N**) *Rpn7* knockdown testis. (**C**,**G**,**K**) Whole images of testes; (**D**,**H**,**L**) Apical regions of fly testes. Germ cells were stained with Vasa antibody (red); Nuclei were stained with DAPI (blue). (**E**) The spermatid nuclei were tightly bundled (white arrows) in the base region of the control testis, but the spermatid nuclei were scattered or disorganized (white arrows) in *Rpn6* knockdown testis (**I**) or were not observed in *Rpn7* knockdown testis (**M**). DAPI staining shows the seminal vesicle full of mature sperms with needle-like nuclei in control testis (**F**) or the seminal vesicles with no sperms in *Rpn6* or *Rpn7* knockdown testes (**J**,**N**). SV: Seminal vesicle. (**D**,**E**,**H**,**I**,**L**,**M**) correspond to the dotted box areas in (**C**,**G**,**K**), respectively. Dotted frames in (**D**,**L**) represent the 16-cell stage of germ cells. Scale bars: 100 μm (**C**,**G**,**K**); 25 μm (**D**,**F**,**H**,**J**,**L**,**N**); 12.5 μm (**E**,**I**,**M**).

**Table 1 ijms-23-09459-t001:** Primers for qRT-PCR and PCR.

Genes	Forward Primer (5′–3′)	Reverse Primer (5′–3′)
*Rpn12R*	GGGCAGATACAACAAGATA	CCTCGTCCAGACACCACT
*Rpn7*	TGCCTTCCGCAAGACCTA	CGCCACCGAGTAAACACC
*Rpt4R*	GGTCGGGGAAATCCTGAAGC	AGTGGTCACATCCAGGGCGA
*Prosα7*	AGGCAGCCAACTACAGACA	GAAGTAACCGAAGGAGGAG
*CSN4*	GGAGACGGGTCAGAAACA	TCCAGGACACGAGCATAA
*Rpn6*	TCACGCACTGAGCAACCT	CTTTGGCGGACAGTAGATAG
*Prosβ1*	GCGGAGTGGTCATTGGAG	TCGTGATAGTTCAGCGAGTAG
*Phb2*	CGGCATCCAGAGCGACAT	GCAGATAGGGCAGGTTCA
*His2Av*	GTGGGTCGCATCCATCGT	CCTCGGCGGTCAGGTATT
*Cul2*	AAGAGTGCGAGGAGAAGT	TGAGATTATCGGGTATGG
*CSN5*	AGGTGATGGGTCTAATGCT	AATGGCTCCTGGTATGTCT
*Hsc70-4*	CCTCGGCGGTCAGGTATT	TGCCGAGCAGGTTGTTGT
*rp49*	CGGTTACGGATCGAACAAGC	CTTGCGCTTCTTGGAGGAGA
*wsp*	TGGTCCAATAAGTGATGAAGAAAC	AAAAATTAAACGCTACTCCA

## Data Availability

The raw MS data and search results of this study are available in iProX (www.iprox.org) with id IPX0004320000. All other data are available in this text and Appendix A.

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
