# Peer review of "Comparative Ubiquitome Analysis Reveals Deubiquitinating Effects Induced by Wolbachia Infection in Drosophila melanogaster"

_ijms, 2022, doi:10.3390/ijms23169459_

Round 1

Reviewer 1 Report

I am OK with the corrections the authors made in the MS and their reply to my comments. The new version of Figure 8 with additional fertility data looks more convincing. The explanation of why pairwise comparison was performed with the use of t-test seems to be sufficient. 

Author Response

Thank you!

Reviewer 2 Report

I thank the authors to perform missing experiments and provide raw data in the interest of doing transparent science.  I have only one minor comment that should be easily addressed -

Since the abstract starts with bringing attention to DUB domain relevance to CI, it is important to highlight at the end of the abstract that the findings of the study are important for male sterility phenotype, however, not for CI. I see authors have stated this information clearly in the discussion, though it would be adequate to highlight in the abstract too.

For example L26-L27 can be rephrased as: "Wolbachia can widely......male fertility of their hosts, but are not involved in CI".

Author Response

We have revised at the end of the Abstract to be: “Wolbachia can widely deubiquitinate proteins that have crucial functions in male fertility of their hosts, but are not involved in CI.” (Page 1, Line 26-27)